# Numerical Analysis of the Seismic Behavior of a Steel Beam-to-Concrete-Filled Steel Tubular Column Connection Using External Diaphragms

**Cristhian Ramírez Ortiz [1], Albio D. Gutierrez Amador [2,*] and Jose L. Ramírez Duque [3]**

[1] School of Civil Engineering and Geomatics Engineering, Universidad del Valle, Cali 760034, Colombia
[2] Research Group in Fatigue and Surfaces, School of Mechanical Engineering, Universidad del Valle, Cali 760034, Colombia
[3] Civil and Industrial Engineering Department, Pontificia Universidad Javeriana Cali, Cali 760031, Colombia
[*] Correspondence: albio.gutierrez@correounivalle.edu.co

**Abstract:** The seismic behavior of a novel steel beam-to-concrete-filled steel tubular column connection with external diaphragms was evaluated numerically by using a model based on the finite element method. The model was validated against experimental results performed in previous work, which revealed that this type of connection is suitable for use in special moment frame structures located in areas of high seismic threat. The comparison included the hysteresis curve, plasticization mechanism, and plastic hinge location. Then, the model was used to study the behavior of the connection for a monotonic test. It was observed that the monotonic test can be used for the qualification of the connection, avoiding the excessive time required for the solution of the cyclic load test. In addition, two stages of simplifications were conducted in the model, showing satisfactory results and significant reductions in computational time. For the first stage, the lateral support beam was removed and replaced by a connection with an infinite lateral rigidity. Second, beam-type elements were implemented in the remote areas of the plastic hinge location. Finally, the simplified model was used in a parametric study that examined the behavior of the connection under four different European type I profiles. It was observed that all the profiles meet the requirements for seismic performance.

**Keywords:** beam-to-column connections; composite structures; cyclic loading; plastic hinge; special moment frames; numerical models





## 1. Introduction

Even though special moment-resistant frames (SMFs) can be considered a recent development in building codes, their use can be traced back to the first reports of the use of structural steel in building construction more than one century ago [1].

At present, there is an increased demand for SMFs in civil construction. This is due to their good dissipation response to seismic events and their versatility in architectural applications. These frames impose smaller forces on foundations than other structural systems, which results in more economical foundation systems. Nonetheless, seismic provisions must be guaranteed in their design. This was widely discussed by different authors, for example, in the work of Andrade [2], on the qualification of steel I-beam connections connected to column weak axis, Bustamante [3], in his work on the qualification of a composite tubular SMF connection, Ceron [4], on the prequalification of a beam-column moment welded connection with dog bone section reduction in the beam, Cheng et al. [5], on the evaluation of the seismic performance of steel beams to concrete-filled steel tubular column connections, Ramirez [6], in his work on the analysis of the inelastic behavior of a double T connection [6], Schneider and Alostaz [7], on the experimental analysis of connections to concrete-filled steel tubes, Sheet et al. [8], on the experimental investigation

of concrete-filled tubular (CFT) column-to-steel beam connections under cyclic loading, Shin et al. [9], in their work on the behavior of welded CFT column to H-beam connections with external stiffeners, Torres [10], on the qualification of a rigid metal connection I beam—composite column, and by Vallejo and Clement [11] on the evaluation a steel beam's rigid connection to a concrete-filled tubular column when submitted to dynamic load.

Since their inception, the methodology for the design and construction of SMFs has been constantly reviewed to consider the requirements of new construction projects [12,13]. This can be observed in the updates to building codes such as AISC 358-16 [14] and AISC 341-16 [15], in which the design philosophy has been updated to prevent failure induced by strong seismic events. In addition, a special focus is placed on the versatility of frames that can be used in multiple civil constructions, and whose design allows for significant damage during a strong earthquake [16].

SMFs include three basic elements in their construction: beams, columns, and beam-column connections. A particular type of SMF uses a concrete-filled steel tubular (CFST) column. This setup is used extensively in architectural and industrial applications, mainly because of its load-bearing capacity under different loading conditions [17]. In addition, the use of diaphragms is common since they provide adequate structural integrity [17,18].

Due to their multiple advantages for construction and assembly, a variety of SMFs are included in building codes, such as AISC 358. Nonetheless, due to the multiple options in the configuration of SMFs, some are not covered under building codes and must be tested according to the provisions included in AISC 341 [15]. This is the case of the connection analyzed in this work, which uses a concrete-filled steel tubular column connection and external diaphragms.

The experimental work presented by Ramírez et al. [19] was aimed at the study and qualification of the connection modeled in this work. For this, an experimental test was conducted in accordance with the FEMA 350 standard for earthquake-resistant structures [12]. The failure modes, hysteretic performance, strength and stiffness degradation, rigidity classification and energy dissipation were determined and analyzed. It was concluded in the experimental study that this connection exhibits large hysteretic loops and develops ductility and dissipation capacity. More importantly, the maximum rotation of the beam was 0.07 rad with a resistant moment above 80% of the beam capacity measured from the face of the column. Thus, the ductility design requirements for earthquake resistance were met according to the current regulations. A review of the experimental work conducted on different configurations of CFST column-to-steel beam connections to analyze their seismic behavior was also conducted by Ramirez et al. [19].

Despite the good results obtained in the experimental tests presented by Ramírez et al. [19] for the steel beam-to-concrete-filled steel tubular column connection using external diaphragms, there are multiple possibilities in the configuration of this type of connection. Testing these configurations will require the development of an expensive and time-consuming experimental test program. A numerical model is a key tool to tackle this issue, i.e., the numerical model can be used to conduct low-cost evaluation of the performance of a particular configuration or group of configurations. Moreover, if such a model can be simplified, it will be a convenient tool to provide fast and satisfactory answers for engineering design purposes.

Examples of the modeling efforts on different configurations of CFSTs can be found in the literature. For instance, Cheng, C.T. and Chung, L. [5] developed an analytical nonlinear force–deformation model to simulate the shear transfer behavior in the panel zone of CFST beam-column connections. In this model, the authors considered the influence of the axial load on the shear transfer behavior. The model was validated with experimental tests conducted on five different types of CFST beam-column connections. It was observed that better ductility of connections occurred at higher axial loads. In addition, the predictions of the model were better for higher axial loads but showed a more conservative behavior for lower axial loads.

Shin et al. [9] presented a numerical model to describe the behavior of CFT columns to H-beam welded moment connections with external T-stiffeners under cyclic loads. The model was based on the finite element method (FEM) and aided by the software ABAQUS. The model was evaluated by comparing the test results for displacement responses and the potential of failure modes. The authors reported good agreement in the prediction of the model. In addition, the model indicated that a properly designed T-stiffener leads to the formation of the plastic hinge in the beam section away from the column face.

Wu et al. [20] developed a mechanical model to describe the theoretical equations to calculate the stiffness, yielding shear strength, and ultimate shear strength of the panel zone for a proposed new design of bolted beam-to-column connections for CFT. The model considers the behavior of steel and concrete elements as independent and accounts for the presence of holes in both steel and concrete elements. The model was validated with a series of cyclic loading experiments that were in close agreement with the experimental results. It was also shown that the proposed connection met the specifications for seismic resistance while presenting good energy dissipation capacity with plastic angular displacements of more than 5%.

Li et al. [21] developed an analytical model to describe the seismic behavior of a connection for circular CFST column-to-steel beam composite structures. The connection was characterized by an extended endplate welded to a steel beam and bolted to a CFT column using high-strength steel rods. The model is based on the FEM and showed good agreement with experiments. The model was implemented using the object-oriented software framework OPENSEES and was validated against experimental tests. The tests showed that the connection exhibited good ductility and energy dissipation capability, meeting the requirements recommended by the AISC.

Tao et al. [22] developed a numerical model based on the FEM to analyze the behavior of bolted end-plate joints for (CFST) columns, steel beams, and through-bolt connections. The model was validated with experimental tests conducted under lateral cyclic loading with horizontal displacements imposed at the top of the column. The proposed model showed satisfactory agreement with the experiments. In particular, the three typical stages, elastic, elastic–plastic and load descending, could be identified from the full range of the load–displacement skeleton curves. The model also captured the buckling effect of the beam flange and web in a satisfactory manner. The model was also used to compare the performance of the bolted joint with that of the counterpart with an external diaphragm.

Xu et al. [23] developed a finite element model using ABAQUS to evaluate the seismic performance of a damage-tolerant steel frame. This type of frame is provided with a composite ultrahigh-performance concrete (UHPC) joint and friction damper applied at the beam-to-column connection. Pushover analysis and nonlinear dynamic analyses were carried out to compare the behavior of the proposed damage-tolerant steel frame against a conventional frame. The model results showed that, compared to the conventional frame, the deformation and the base shear force of the novel frame are significantly reduced. It was also concluded that the early yielding mechanism caused by the weak friction dampers can effectively improve the energy dissipation performance and damage control.

Wu et al. [24] developed a finite element model in ABAQUS to describe the seismic performance of a steel-reinforced concrete column-steel beam composite joint (MPCJ). They analyzed three different beam-column connection types: bolted, welded, and bolted-welded. The connections were subjected to low-cycle reversed loading to investigate the elastic and elastoplastic development trends, failure characteristics, and seismic response. Based on the experimental results, it was concluded that the MPCJs exhibited stable hysteretic curves, reasonable strength and stiffness degradation and good ductility and energy dissipation performance. These results were used to validate the finite element model, which showed good agreement with the experiments. In addition, based on the experimental results and the numerical validation, the authors proposed simplified equations to calculate the flexural and shear-bearing capacity of MPCJs.

Rong et al. [25] conducted an FEM-based analysis of the seismic performance of a steel frame with an external diaphragm joint between a CFST column and an H-shaped steel beam. The numerical model considered material and geometric nonlinearity. The model was validated with the hysteretic and skeleton curves obtained for a quasistatic test conducted to analyze the joint stiffness, beam-to-column stiffness ratio and concrete strength. Although the model was in good agreement with experiments, the authors reported that the numerical model had a plumper hysteretic curve and was more rigid than the experimental results. They attributed this to the initial defects of the material being ignored and the simplification of the weld. In addition, the authors reported that the model was in good agreement for the deformation and stress distribution.

Mou et al. [26] conducted a numerical analysis to evaluate the seismic performance of a novel connection between a beam and a reinforced concrete-filled steel tube (RCFST) column. The connection was tested under cyclic loading to evaluate the failure modes, hysteretic performance, stiffness degradation, strength degradation, energy dissipation capacity, and strain responses. The authors compared the skeleton curves of the experiments and the numerical model. It was observed that the curves were in good agreement before the peak load. They attributed the mismatch beyond the peak load to the difficulty in simulating the behavior of bond-slip between steel parts and concrete parts and the cracking behavior of concrete. The authors also reported that the failure modes predicted by the model and observed experimentally were similar.

Li et al. [27] proposed a numerical model based on the FEM to study the seismic performance of a novel U-shaped diaphragm connection designed to transfer the moment at beam ends in the frame with special-shaped CFST columns and steel beams. The model was verified by comparing the horizontal load-interstory drift hysteretic curves of the tests. Then, the model was used to analyze the influences of the U-shaped diaphragm size, tube thickness, and axial load ratio of the column. Based on the parametric analysis results, a mechanical model was proposed to calculate the yield strength and ultimate strength of U-shaped diaphragm connections. The authors reported good agreement between the strengths calculated by the mechanical model and the FEM-based model.

Previous research in the field highlights the relevance of developing numerical models that support the design and analysis of a particular portfolio of connections. Nonetheless, none of the models available in the literature can be used to study the seismic behavior of the steel beam-to-concrete-filled steel tubular column connection using external diaphragms qualified experimentally by Ramírez et al. [19]. In this context, we present a FEM-based model and its application to predict the seismic response and behavior of this connection. This is presented showing relevant aspects of the numerical model for the design of this type of connection in terms of its flexural capacity, plasticization mechanism and plastic hinging location. The analysis was conducted according to the criteria established by current regulations and the results of previous experimental tests. In addition, two stages of simplifications were conducted in the original model aimed at improving its suitability for engineering purposes.

## 2. Materials and Methods

### 2.1. Description of the Connection

The design procedure, geometry manufacturing process and assembly of the connection are presented in detail in the literature [19]. Figure 1 shows the components and main dimensions of the connection. The action of the maximum capacity force $F_{pr}$ and the maximum probable moment $M_f$ are also illustrated. As shown in the figure, the column is made of ASTM A500 grade C steel and has a circular section with an outer diameter of 323.9 mm and a nominal wall thickness of 9 mm. The beam corresponds to an IPE360 rolled profile and is made of ASTM A572 grade 50 steel. The actual strengths of the steel were estimated according to the ASTM E8-11 standard [28]. For the column, the yield and ultimate limit of the material were 410 MPa and 563 MPa, respectively, whereas for the beam, the corresponding values were 386 MPa and 540 MPa. The column was filled with

concrete with a compressive strength of 21 MPa. The diaphragm thickness was 19 mm, with 16 bolts per diaphragm (bolt diameter of 5/8 inches (15.875 mm)), and was made of ASTM A490 steel. Diaphragms were connected to the column wall using full penetration welds with 45-degree bevels. Table 1 shows the relevant values of the connection.

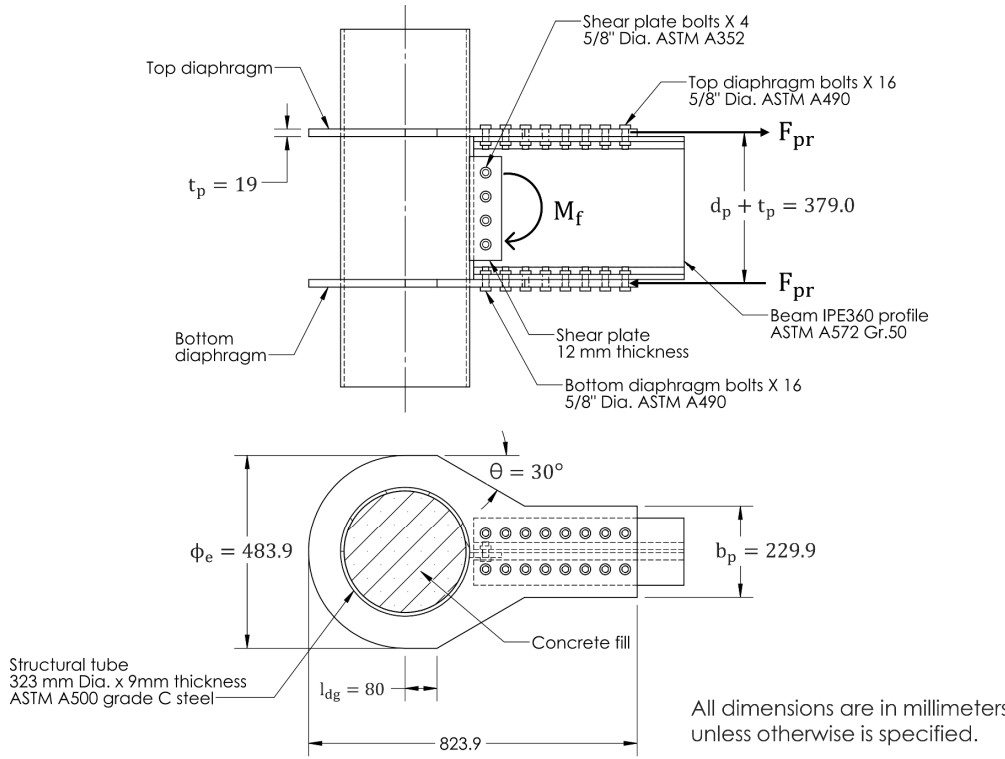

**Figure 1.** Components and main dimensions of the connection obtained from the design process. The action of the maximum capacity force $F_{pr}$ and the maximum probable moment $M_f$ are also illustrated [19].

**Table 1.** Relevant parameters required for the design of the beam-column connections analyzed in the parametric study.

| | | | **Value for Each Profile** | | | |
|---|---|---|---|---|---|---|
| **Notation** | **Description** | **Unit** | **IPE 330** | **IPE 400** | **IPE 450** | **IPE 500** |
| $d_b$ | Beam profile web length | mm | 330 | 400 | 450 | 500 |
| $d_c$ | Column outer diameter | mm | 273.1 | 323.9 | 377.9 | 377.9 |
| $t_{cw}$ | Column wall thickness | mm | 9.3 | 10 | 11.1 | 11.1 |
| $\phi_e$ | Diaphragm outer diameter | mm | 407 | 484 | 563 | 563 |
| $t_p$ | Diaphragm thickness | mm | 15 | 19 | 22 | 25 |
| $d_{bolt}$ | Bolt diameter | mm | 15.88 | 15.88 | 19.05 | 19.05 |
| | | (in) | (5/8) | (5/8) | (3/4) | (3/4) |
| $N_b$ | Number of bolts | - | 12 | 16 | 14 | 14 |
| $Z_x$ | Plastic module of the beam | cm$^3$ | 804 | 1307 | 1702 | 2194 |
| $f'_c$ | Concrete strength | MPa | 21 | 21 | 21 | 42 |
| $F_{yb}$ | Beam yield strength | MPa | 350 | 350 | 350 | 350 |
| $E$ | Modulus of elasticity of steel | MPa | 200,000 | 200,000 | 200,000 | 200,000 |

### 2.2. Description of the Experimental Procedure

The experimental setup, including the mounting details of the beam-column connection, is discussed in detail in the literature [19]. The test was carried out at the homologation framework lab (with the acronym MaPH in Spanish) located at the School of Civil Engineering of the University of Valle. Figure 2 shows the experimental setup, including the

mounting details of the beam–column connection. As shown in Figure 2a, two supports were installed at each end of the column under the following constraint conditions: at end 1, translations in the plane and out of the plane were restricted; at end 2, vertical translation and translation outside the plane were also restricted and rotation in the plane was allowed for both ends. Figure 2b illustrates the beams that provide lateral support to the framework.

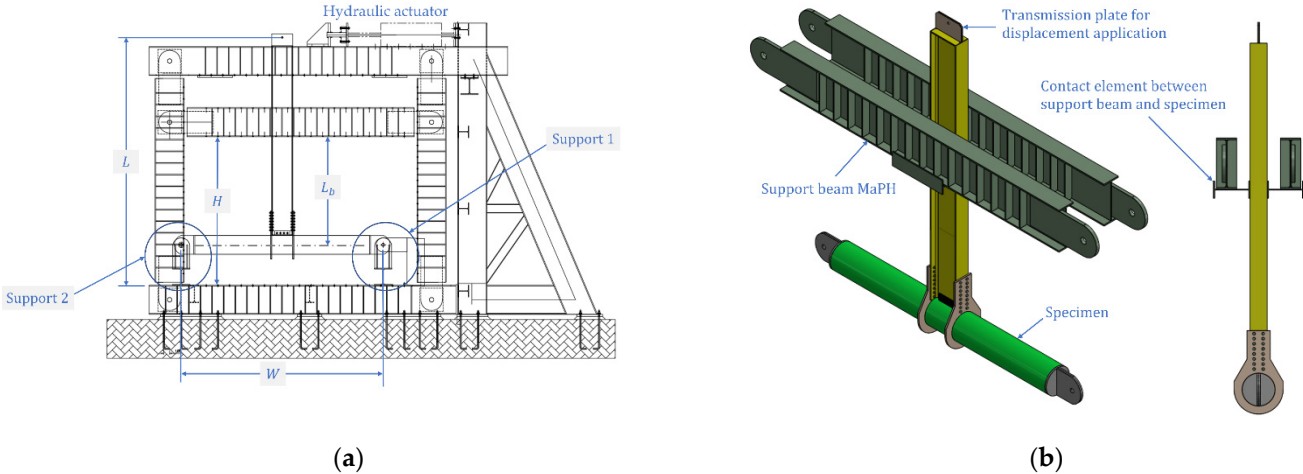

(**a**)                                                                                          (**b**)

**Figure 2.** Experimental setup: (**a**) mounting details of the beam-column connection; and (**b**) beams that provide lateral support to the framework [19].

Figure 3 shows the protocol applied for the test. The displacement protocol was programmed according to the procedure described in AISC 358 [14]. This protocol corresponds to linear cyclic displacements divided into 11 discrete steps. The test allowed us to obtain the hysteresis curves, the mechanism of plasticization and the form of failure if such failure occurred.

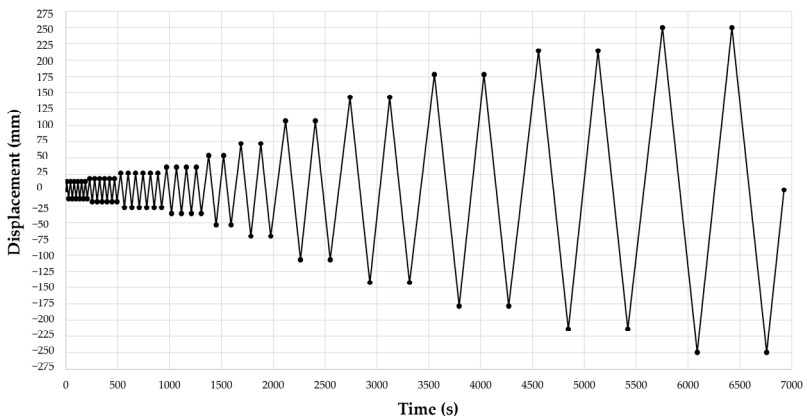

**Figure 3.** Displacement protocol. The abscissa is presented in units of time in s, obtained as the quotient between the displacement $\Delta$ and the loading application speed $v$. In this case, $v = 1.49$ mm/s [19].

### 2.3. Description of the Model

To appropriately capture the complexity of the problem, the model requires the definition of the geometry domain, including its appropriate discretization in finite elements, the constitutive properties of the materials and an appropriate set of boundary conditions. The boundary conditions should include the characteristics of the supports, the contact between elements and application of the displacement protocol.

### 2.3.1. Geometry Domain

The geometry domain was built to accurately represent the system as well as the configuration and dimensions of the test. The geometry was divided considering the 49 bodies that compose the connection: 32 bolts, four beam sections, three tubular column sections, three concrete volume sections, two diaphragms, two end caps in columns, two plates for column supports, and one plate for the application of the displacement protocol. These divisions are shown in Figure 4. The contact surfaces between the diaphragms and the column, cutting plate—column, end caps—column and support plates—end caps were considered bonded. In addition, some elements of the connection, such as nuts and bolts, were simplified as cylindrical elements to avoid excessive distortions in the edges of the hexagon that the real geometry describes.

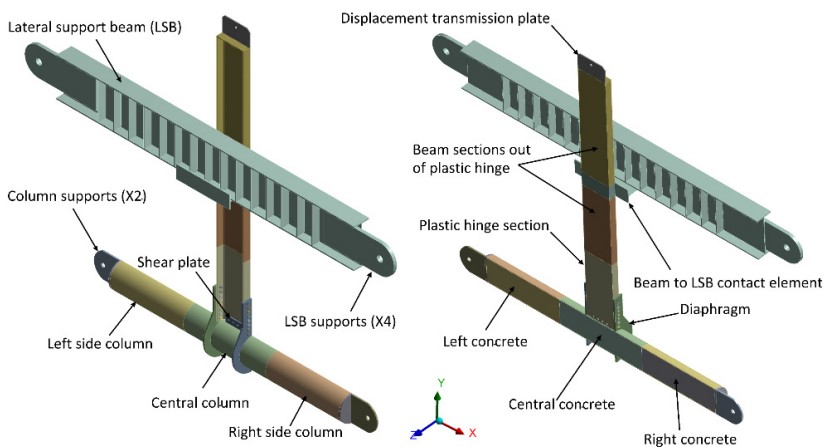

**Figure 4.** Part division, illustrating the plane of symmetry.

Due to the symmetry of the system regarding the geometry, loads and constraints, it was only possible to consider half of the domain, with the XY plane as the symmetry plane. Figure 4 shows the part division of the model organization.

### 2.3.2. Domain Discretization

The domain discretization in finite elements was performed according to the division explained above. For the selection of the finite element types, three main effects were considered. The first is the strain hardening of the steel. For this, a plasticity model was included to account for the fact that the steel, after reaching its yield point, is capable of deforming in the plastic range without degradation of its resistance. The second aspect is the contact effect. This phenomenon may be characterized by separation between the surfaces of the connection elements. In this case, separation occurs in the diaphragms that are initially in contact with the beam flanges as the cyclic displacement protocol advances. This is due to the dissipation of the friction component. The third effect considered is the nonlinearity induced by the large deformations that occur in the connection in response to the applied forces.

Based on these considerations, the element defined in the Ansys software as SOLID186 [29] was selected. SOLID186 is a higher-order three-dimensional element that exhibits quadratic displacement behavior. The element is defined by 20 nodes with three degrees of freedom per node and translations in the x, y, z nodal directions. This type of element allows for modeling the plasticity of the material and the effects of large deformations.

Figure 5 shows the domain discretization as well as a closer look at the regions where the mesh was refined the most, i.e., the left and right ends of the beam, the beam outside the plastic hinging and the lateral support beams.

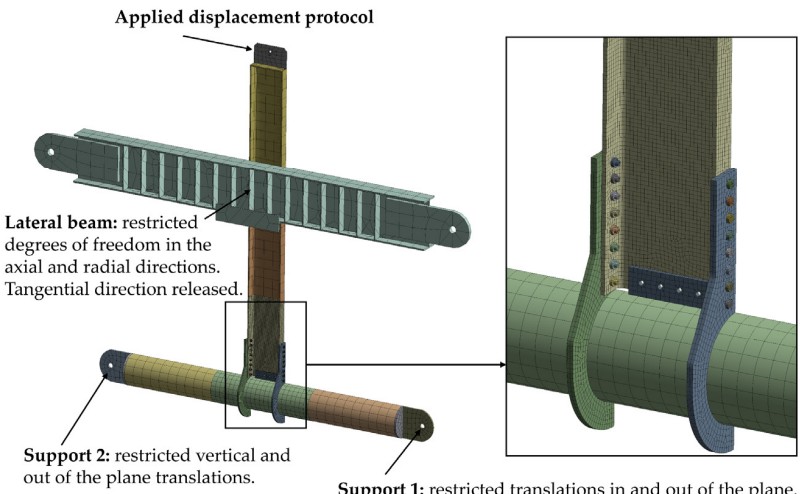

**Figure 5.** Domain discretization.

### 2.3.3. Boundary Conditions

The boundary conditions defined in the model are the displacement protocol, supports, contacts, and bolt prestressing. The displacement protocol was discretized in 72 load steps and applied in the displacement application zone (see Figure 5). The constraints at the ends of the column are also shown in Figure 5. In support 1, translations in and out of the plane were restricted; in support 2, vertical translation and out-of-plane translation were restricted; however, in-plane rotation was allowed for both supports. In the lateral support beam, the degrees of freedom were restricted in the axial and radial directions; however, the tangential direction was released (see Figure 5).

Three types of contacts were implemented in the model: adhered, without friction and with friction. Bonded-type contacts were used for the diaphragm-column contact pair, the pairs of contacts that were made to have a geometric division for the discretization in the mesh, the contact pairs between the plates of the column supports, and the contact pairs between the displacement transmission plate and the beam outside the plastic hinge. The frictionless contact type was implemented to consider the phenomenon of deformation that occurs in the holes by crushing due to the interaction of the bolts with the beam and the diaphragms. This phenomenon was studied and understood from the numerical and experimental research carried out by Kim and Kuwamura [30]. Finally, the frictional contact type was applied to the contact pairs between the diaphragms and the flange of the beam, the diaphragms and bolt heads, and the nuts and the flange of the beam.

To simulate the prestressing in the bolts, an element with a single degree of freedom was used, which represents the direction defined for the prestressing.

Notably, the contact surfaces between the diaphragms and the beam flanges were joined by considering no relative displacement between them during the first cycles. This is due to the friction effect achieved by prestressing the bolts. Additionally, it is important to consider that this friction is determined by the normal force transmitted by the bolt and the friction coefficient between the contact surfaces. In this work, the friction coefficient considered in the model was 0.19, according to the recommendations provided by the AISC standard [31].

Another important effect occurs with the advancement of the displacement cycles. In this case, a bending moment develops in the connection. If the decomposition of the bending moment into a pair of statically equivalent forces is considered and the magnitude of these forces is greater than the friction force, the plates begin to slide, causing the bolts to experience shear forces.

### 2.3.4. Material Model

To account for the large inelastic deformations that occur during the plasticization process of the beam, the nonlinear behavior of the material was considered in the model. For this, the kinematic strain hardening component was included according to the results obtained by Andrade [2], where the size of the yield surface remains constant and moves in the direction of the load. In Figure 6, the multilinear curve is defined by kinematic hardening. In this curve, the vertical axis shows the stresses in MPa, and the horizontal axis shows the component of plastic deformation in micrometers.

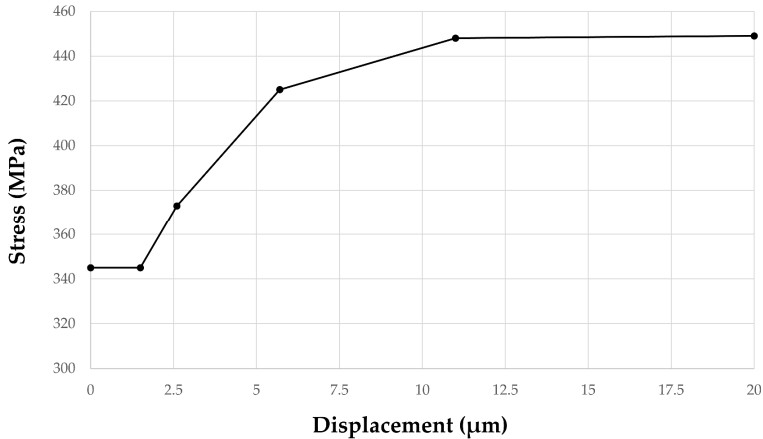

**Figure 6.** Multilinear curve of kinematic hardening.

### 2.3.5. Model Simplification

Two stages of model simplification were performed to reduce the computational time required to analyze the behavior of the system. For the first stage, the lateral support beam was removed to verify the behavior of the connection with infinite lateral rigidity. This was made by setting the out-of-plane displacement condition equal to zero to reproduce the optimal condition recommended by the AISC 360 standard [16]. Figure 7a shows the division made to the beam to provide lateral support for the strip where the condition of zero displacement was applied. The results obtained from this simplification are discussed in Section 3.

For the second stage, the key idea was to use beam-type elements in the remote areas of the plastic hinge location. These beam-type elements were split into two parts, with the geometric location of the split corresponding to the distance where the lateral support was located. This simplification was performed while maintaining nodal compatibility between the solid elements and the beam-type elements.

The beam-type element selected was BEAM 188 [29]. This element has two node ends. At each end, the model has six degrees of freedom, three displacements and three rotations. In addition, it includes the geometric properties of the cross section.

The coupling between the SOLID 186 and BEAM 188 elements was performed using MPC184 elements. The MPC184 element can provide compatibility through constraint equations applied to the nodes of the surface of the SOLID 186 element with the end of the BEAM188 element [29]. Figure 7b illustrates the model simplification.

Additional simplifications to the model in the second stage were performed as follows: (i) displacement protocol: the monotonic displacement protocol was applied on the transmission plate, which was modeled as a solid element; (ii) supports: the supports were implemented through solid elements; (iii) contacts: the contact between the diaphragm and the beam flange was the frictional type; (iv) bolts: the bolts were modeled as beam-type elements considering the circular geometry of the bolt and the connectivity with the holes of the diaphragm; (v) prestress: prestress was included for the bolts modeled as beam-type elements.

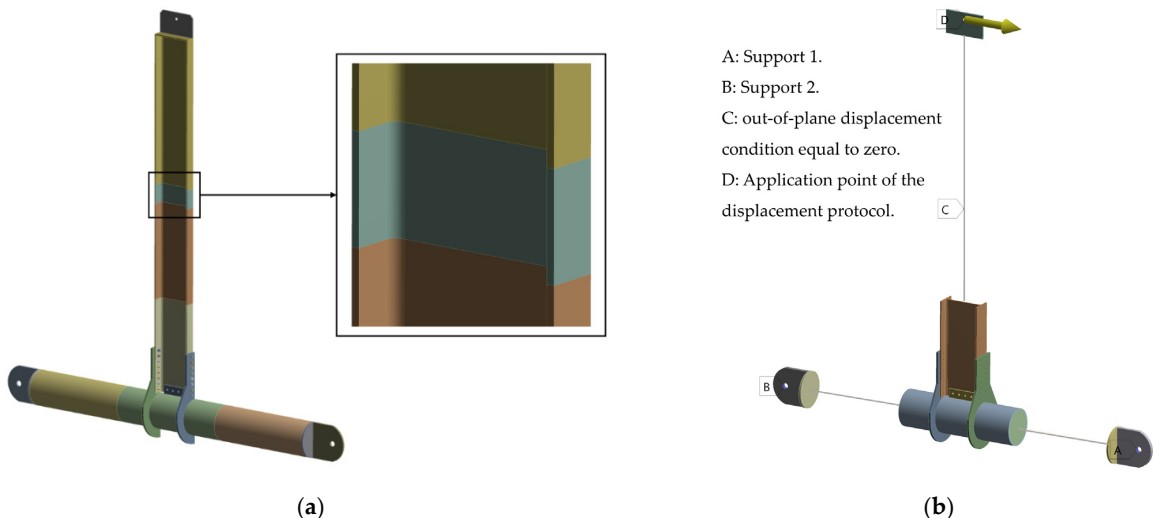

(a)          (b)

**Figure 7.** (**a**) Applied condition for the first simplification of the model; (**b**) Model simplification using SOLID186, BEAM 188 and MPC184 elements. The boundary conditions are also shown.

## 3. Results and Discussion

### 3.1. Description of the Connection

Figure 8 shows the results obtained for the hysteretic curve of the numerical model. In this figure, the moment is described as a function of the angle of rotation. It can be observed that the resistance of the connection at 0.04 radians is 432.3 kN-m, which is higher than the estimated value at 0.8 MP, where MP is the plasticization moment of the beam.

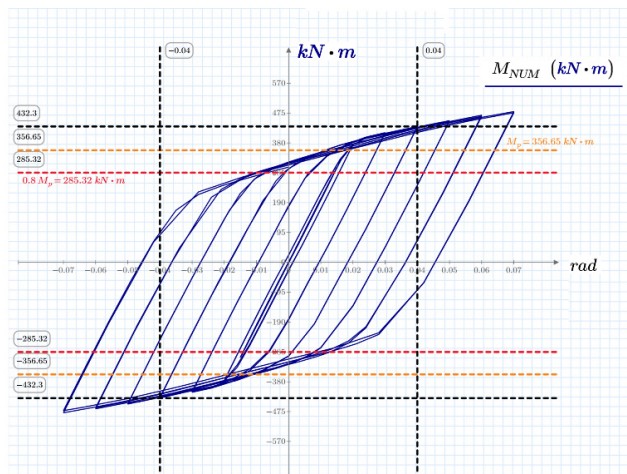

**Figure 8.** Hysteretic curve (moment–angle).

Figure 9 shows the comparison between the hysteretic curves obtained numerically and experimentally [19]. The numerical model shows a hysteresis curve with a rigidity during the loading and unloading cycles similar to that observed experimentally. It can also be observed that although the nonlinear effect of the connection was captured, it does not include the pinching effect due to bolt slip because the space between the bolt and the hole was not considered, i.e., due to the clearance of 3 mm between the bolt and the diameter of the hole, there is relative rigid body movement between them. The bolt will move, making intermittent contact with the hole surface at different points with the associated contact stresses and strains, including the dissipation of kinetic energy. This effect, in addition to the geometric nonlinearity of the problem due to large deformations and the nonlinearity of the material when exceeding the elastic range, complicates the problem excessively.

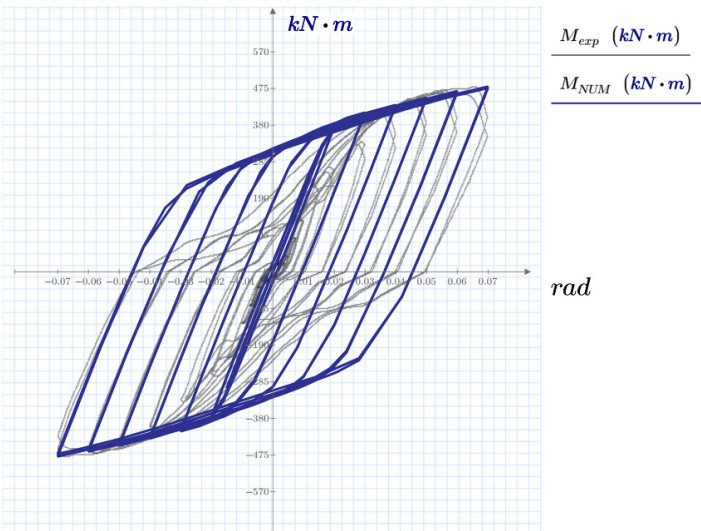

**Figure 9.** Experimental and numerical moment-rotation curves.

To address the nonlinearity caused by the pinching effect, a practical approach was implemented aimed at a good compromise between accuracy and the demand for computational resources. Thus, a fitting coefficient of 0.9 was considered to account for the pinching effect due to bolt slip, bearing in mind that the moment ordinates in the hysteretic curve obtained experimentally would be higher than those in the numerical model due to the phenomenon of rigid body movement and intermittent contact between the bolt and the hole.

### 3.2. Plastification Mechanism

Experimentally, the plastic hinge location was observed at a distance $S_{h\_exp}$ = 480 mm. At this location, an important concentration of stresses and large inelastic deformations occur. However, the plastic hinge location predicted by the numerical model has a value of $S_{h\_NUM}$ = 480 mm, i.e., the numerical model presents a deviation of only 2.5% with respect to the experiments. Figure 10 shows the comparison between the plastic hinge location predicted by the numerical model and the experiments conducted in previous work [19].

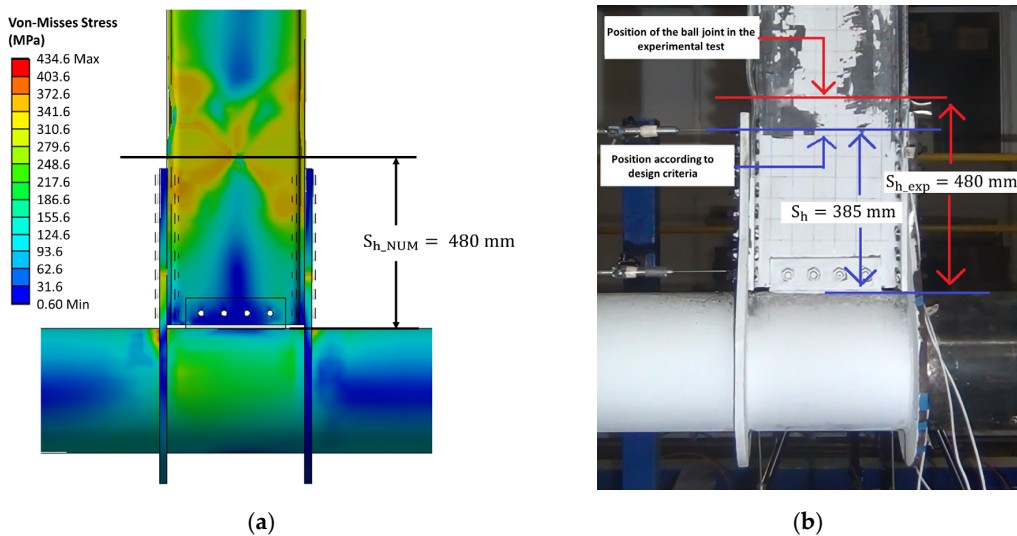

(**a**)                                                            (**b**)

**Figure 10.** (**a**) plastic hinge location and (**b**) comparison with the observations obtained experimentally [19].

### 3.3. Monotonic Test

According to FEMA 355D [13], monotonic curves provide an approximate envelope to those obtained from hysteretic curves in cyclical load tests. Figure 11 presents a comparison between the numerical and experimental envelopes, showing a good correlation between the angular rotation from 0.04 to 0.07 rad.

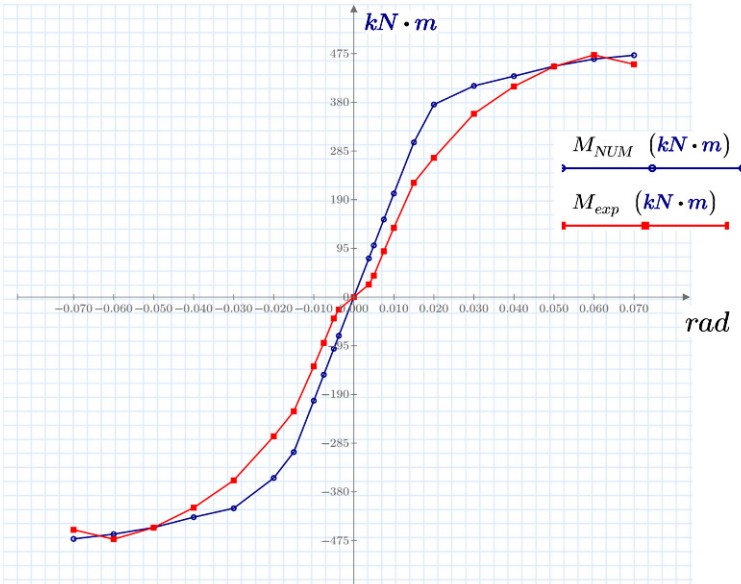

**Figure 11.** Numerical envelope of the moment–angle curve.

The angular rotation between 0.005 and 0.02 rad corresponds to the linear component of the system, and its slope represents the stiffness of the connection. The elastic stiffness of the connection is important during the structural analysis process of moment-resistant gantry systems since lateral displacements and design stresses depend on the capacity of the gantries to develop moment at the beam-column nodes. Thus, the stiffness of both curves was obtained. The experimental curve stiffness was 18,132.5 kN-m/rad, and the numerical rigidity was 20,201.7 kN-m/rad.

Figure 12 presents the moment-rotation curve obtained by the numerical model. For this, a monotonic displacement protocol was applied where the rotation increased from zero to the rotation where the degradation by resistance without discharge occurs. The graph shows the superposition of three curves, namely, the numerical cyclic envelope, the experimental cyclic envelope, and the monotonic response. It is observed that the monotonic response matches the cyclical curve of the numerical model up to 0.06 rad. Beyond 0.07 rad, there is a divergence in the results, given that loss in resistance is not observed, i.e., the resistance moment of the connection does not degrade with the progress of the rotation.

Since the monotonic curve provides a good approximation to the hysteretic behavior up to 0.06 radians, this approximation can be used to decrease the computational time approximately 21-fold, considerably reducing the demand for hardware resources. In addition, this analysis provides a good estimate for the plasticization mechanism and the location of the plastic hinging (see Figure 13), where the deformed position of the connection is shown at rotation angles of 0.04, 0.07 and 0.11 rad.

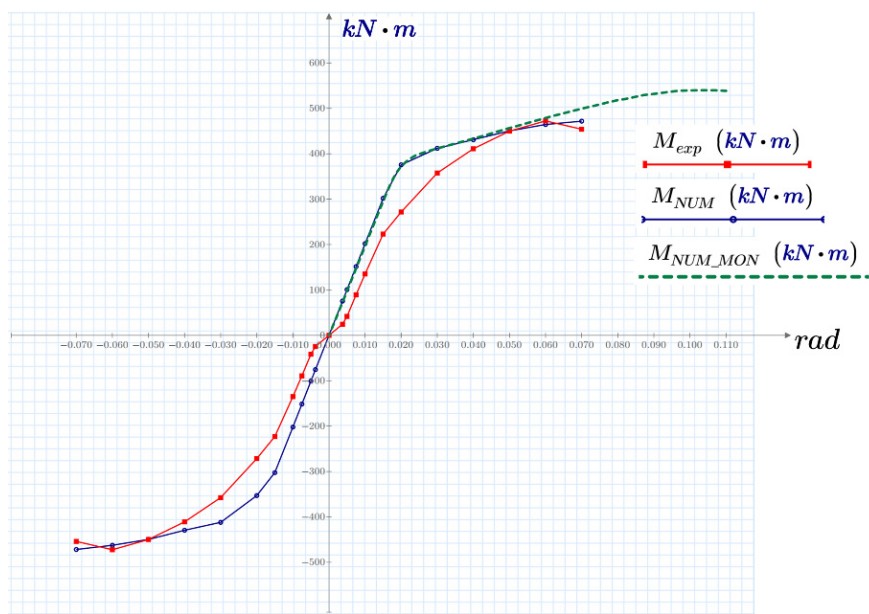

**Figure 12.** Moment–angle curve obtained by the numerical model under a monotonic displacement protocol ($M_{NUM\_MON}$). The curve is compared against the numerical ($M_{NUM}$) and experimental cyclical envelopes ($M_{exp}$).

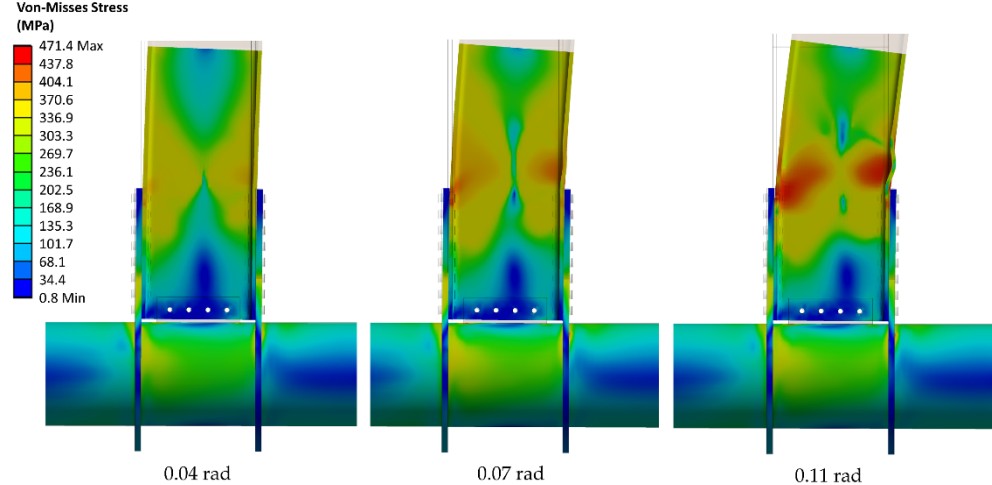

**Figure 13.** Plasticization mechanism exhibited by the numerical model under monotonic displacement.

### 3.4. Model Simplification in the First Stage

Figure 14 presents a comparison between the envelope curves obtained for the experiments ($M_{exp}$), the original numerical model ($M_{NUM}$) and the numerical model after the first simplification ($M_{NUM\_S1}$), in which the lateral support beam was removed to verify the behavior of the connection with infinite lateral rigidity. It is observed that the monotonic test curve represents a behavior similar to the cyclical and experimental curve in the linear elastic part. After a rotation angle of 0.015 rad, there is a difference between the results due to the simplification; however, it is still a good representation of he studied phenomenon, and requires half the computational time.

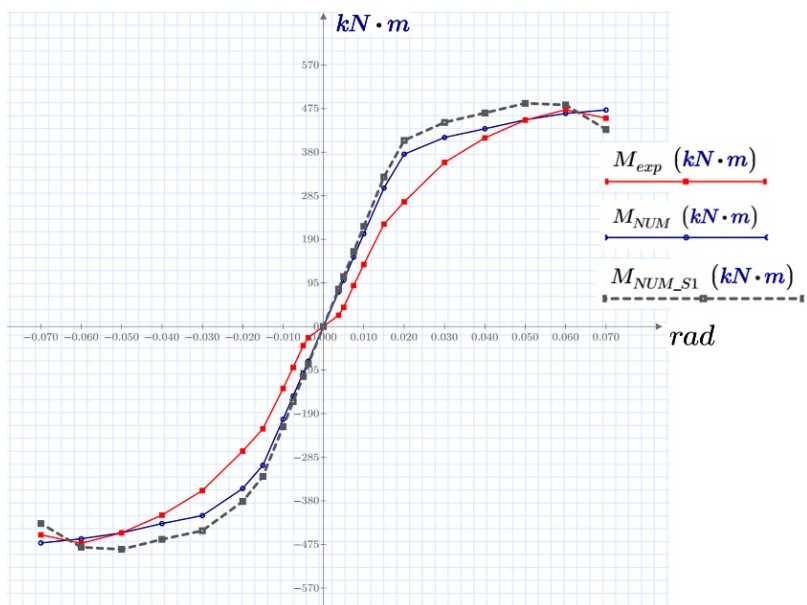

**Figure 14.** Comparison between the envelope curves obtained for the experiments ($M_{exp}$), the numerical model defined originally ($M_{NUM}$) and the numerical model after the first simplification ($M_{NUM\_S1}$).

Figure 15 shows the results obtained for the plasticization mechanism of the numerical model after the first simplification. The deformed position of the connection is shown at rotation angles of 0.04, 0.07, and 0.11 rad. The plastic hinge location and the failure mode can be observed in the figure.

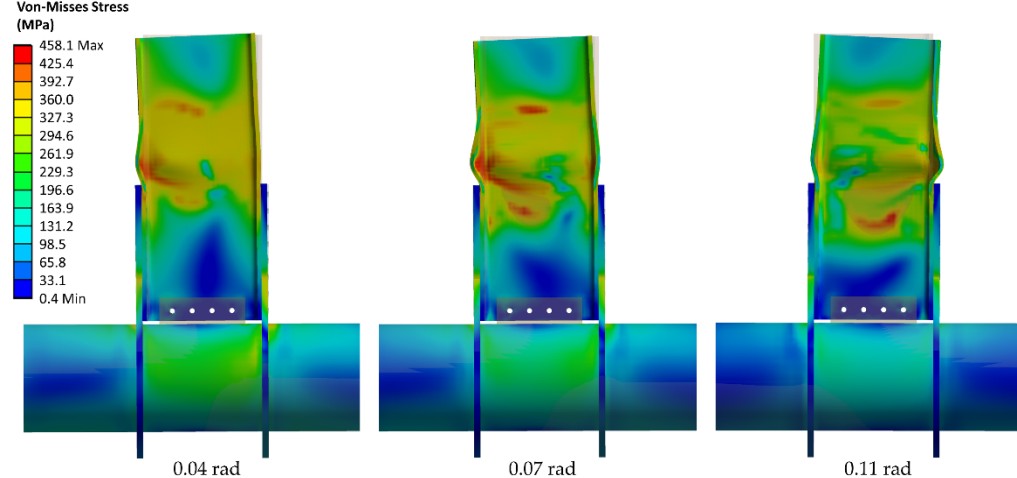

**Figure 15.** Results obtained for the plasticization mechanism of the numerical model after the second simplification.

### 3.5. Model Simplification in the Second Stage

Figure 16 presents a comparison between the envelope curves obtained for the experiments ($M_{exp}$), the original numerical model ($M_{NUM}$) and the numerical model after the simplification performed in the second stage ($M_{NUM\_S2}$).

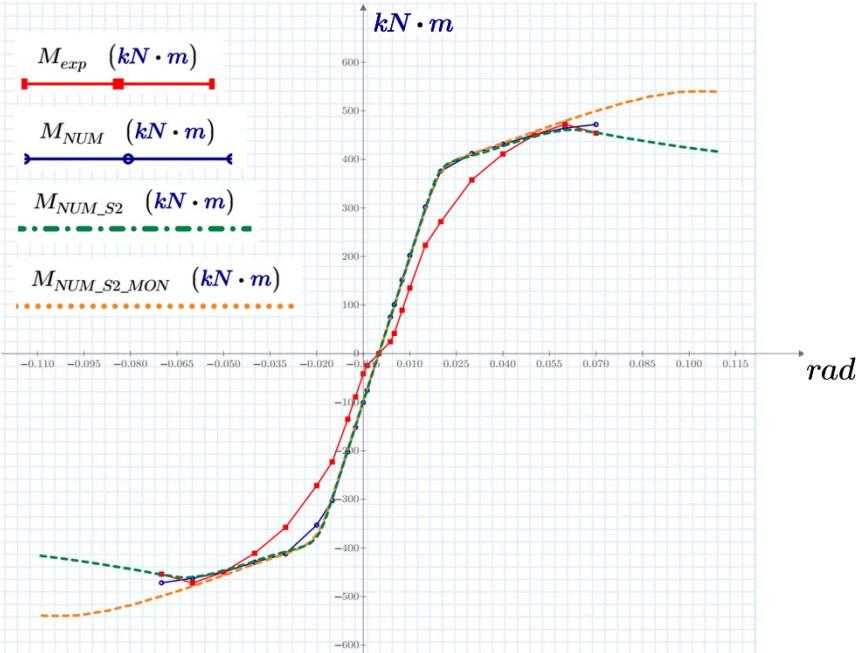

**Figure 16.** Comparison between the envelope curves obtained in the experiments ($M_{exp}$), the original numerical model ($M_{NUM}$) and the numerical model after the second simplification ($M_{NUM\_S2}$).

For the numerical model obtained after the second simplification, the analysis was raised to a rotation level of 0.11 radians to consider the resistance degradation trend due to local buckling in the beam flanges. The green curve in Figure 16 describes this behavior, which corresponds to what is expected once the buckling effect occurs. The local buckling failure is shown in Figure 17, demonstrating that the results provide a good approximation to the actual behavior.

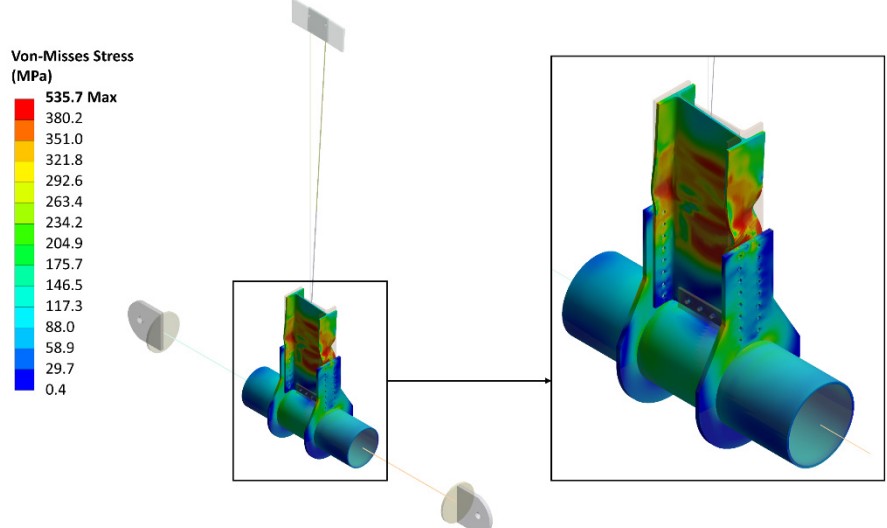

**Figure 17.** Local buckling effect predicted by the numerical model after the second-stage simplifications.

It is worth mentioning that the simplifications conducted in the first and second stages decreased the computational time 56-fold compared with the original model. In addition, if the monotonic test is used instead of the cyclic test, the time can be reduced 84-fold.

### 3.6. Parametric Study

A parametric study was conducted to analyze the seismic response of four different European type I profiles. For this purpose, the simplified model was used due to the good compromise between the accuracy and computational time. The profiles were selected by considering the steel sections with the best performance at meeting drifts and lateral displacement in frames as well as the proper energy dissipation capacity for medium- and low-rise buildings. In all the combinations considered, the strong column-to-weak beam ratio was guaranteed. In addition, the quality and pretension of the bolts and the welding specifications met the requirements of the regulations established by FEMA 350 [12]. Table 1 shows the relevant parameters required for the design of the beam-column connections analyzed in the parametric study.

The corresponding moment–angle curve for each of the configurations analyzed is shown in Figure 18. All the curves exhibit the same resistance degradation trend. The figure also shows the resistance moment of the connection, which was obtained by drawing a line parallel to the ordinate at 0.04 radians. The results are shown in Table 2, and were compared with the limit that must be reached to determine the applicability in special energy dissipation systems. For all the connections analyzed, their observed resistance moment was greater than this limit. Table 2 also shows the plastic hinge location. For all configurations, the plastic hinge was located out of the diaphragm zone.

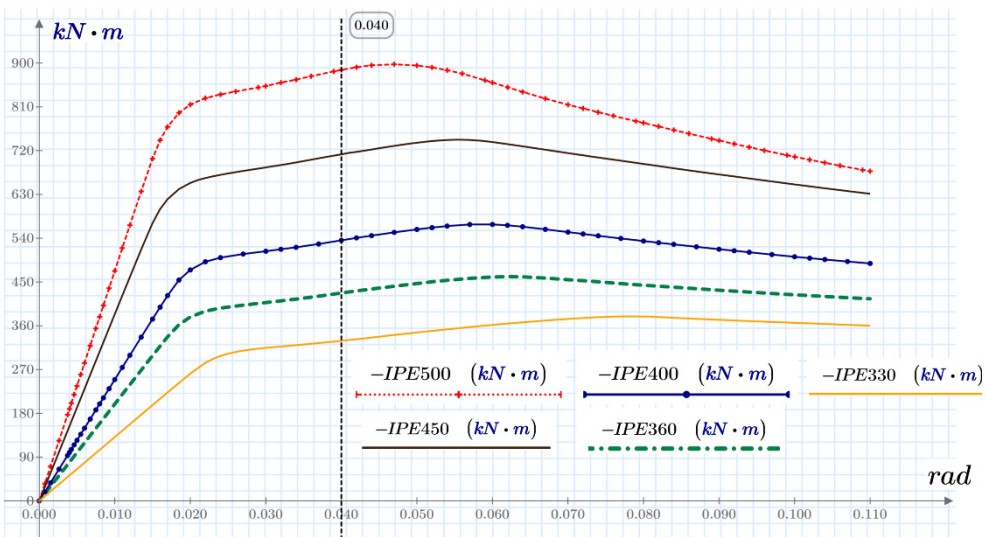

**Figure 18.** Moment–angle curve for each analyzed configuration.

**Table 2.** Results of the parametric study.

| Profile | 0.8 Mp (kN-m) | Resistant Moment (kN-m) | Plastic Hinge Location (mm) | Qualification Verdict (Yes/No) |
|---|---|---|---|---|
| IPE 330 | 224.5 | 328.5 | 403 | yes |
| IPE 400 | 365.7 | 535.4 | 518 | yes |
| IPE 450 | 476.6 | 711.9 | 526 | yes |
| IPE 500 | 613.8 | 885.5 | 542 | yes |

## 4. Conclusions

The increase in the use of SMFs motivates the use of new connections that are not homologated under the current design regulations. Therefore, studies are required to ensure that the designed frames dissipate the seismic energy in the beams without affecting the connections. In this context, this work presents a numerical model and an analysis of the inelastic behavior under cyclic loads of a steel beam-to-concrete-filled steel tubular column connection using external diaphragms.

The model was based on the FEM and built in Ansys, and showed satisfactory results when compared with experiments. In addition, two stages of simplification were conducted to reduce the computational time required for the calculations.

Although the nonlinear effect of the connection considered in the model included the effect of large deformations and the kinematic hardening of the material, the pinching effect due to bolt slip was not included. Nonetheless, the nonlinearity caused by this effect was accounted for in a practical way by a fitting coefficient. This approach led to good agreement with the experimental results in the nonlinear range. In addition, the nonlinear effect of the connection was captured with good results in terms of the location of the plastic hinging and the plasticization mechanism.

It was also observed that the approximate envelope of the hysteretic curves provided by the monotonic test was verified as stated by FEMA 355D [13]. Moreover, a good correlation was observed, especially between the angular rotation from 0.04 to 0.07 rad, where the agreement was within 1%. In addition, the monotonic analysis provided a good estimate for the plasticization mechanism and the location of the plastic hinge. This was also the case after the model simplifications conducted in the first and second stages.

Regarding the reductions in the computational time, it was observed that the use of the monotonic test instead of cyclic loading decreased the time spent to reach a solution by 21 times. The first- and second-stage simplifications decreased the computational time 56-fold with respect to the original model. Finally, if the monotonic test is used along with the simplifications conducted in the first and second stages, the time reduction is 84-fold.

A parametric study was carried out with the numerical model including all the simplifications. In this study, four European type I profiles were analyzed using the monotonic test and the two model simplification stages. To select these profiles, the possible steel sections with the greatest possibility of meeting drifts or lateral displacement in frames with special energy dissipation capacity for medium- and low-rise buildings were considered. The results show that for all the configurations, the resistance moment was greater than the limit that must be reached to determine its applicability in special energy dissipation systems. Thus, all the connections satisfy the qualification requirements of AISC 358 to be used in SMFs with special energy dissipation capacity. In addition, the plastic hinging location is out of the diaphragm zone.

Subsequent studies can be oriented to increase the portfolio of the steel beam-to-concrete-filled steel tubular column connection using external diaphragms by analyzing different configurations, i.e., column sections, beam profiles, and different type of concretes suitable for seismic resistant applications. In future research, the pinching effect can be also studied in more detail. In addition, in this work, the diaphragms were connected to the column wall by using full penetration welds with 45-degree bevels. Nonetheless, it would be worth considering the effect of combining full penetration and partial penetration welds. This, since the use of partial penetration welds would considerably reduce the manufacturing time and associated costs.

**Author Contributions:** Conceptualization, C.R.O., A.D.G.A. and J.L.R.D.; methodology, C.R.O., A.D.G.A. and J.L.R.D.; software, C.R.O.; validation, C.R.O., A.D.G.A. and J.L.R.D.; formal analysis, C.R.O., A.D.G.A. and J.L.R.D.; investigation, C.R.O., A.D.G.A. and J.L.R.D.; resources, C.R.O., A.D.G.A. and J.L.R.D.; data curation, C.R.O. and A.D.G.A.; writing—original draft preparation, C.R.O. and A.D.G.A.; writing—review and editing, A.D.G.A.; visualization, C.R.O. and A.D.G.A.; supervision, C.R.O. and A.D.G.A.; project administration, A.D.G.A.; funding acquisition C.R.O., A.D.G.A. and J.L.R.D. All authors have read and agreed to the published version of the manuscript.

**Funding:** This research was funded by the Universidad del Valle.

**Informed Consent Statement:** Not applicable.

**Data Availability Statement:** The data used to support the findings of this study are available from the corresponding author upon request.

**Acknowledgments:** We thank the Universidad del Valle for supporting this research through the Internal Call for Presentation of Research Projects and Artistic Creation in Sciences, Arts, Humanities, Technologies and Innovation 2019 Award—Project ID 21090.

**Conflicts of Interest:** The authors declare no conflict of interest. The funders had no role in the design of the study; in the collection, analyses, or interpretation of data; in the writing of the manuscript, or in the decision to publish the results.

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
