# Peer review of "Numerical Analysis of the Seismic Behavior of a Steel Beam-to-Concrete-Filled Steel Tubular Column Connection Using External Diaphragms"

_buildings, doi:10.3390/buildings12081217_

Round 1
Reviewer 1 Report
The authors study the seismic behavior of a steel beam-to-concrete-filled steel tubular column connection with external diaphragms. In particular, starting from the experimental tests performed by the same authors, the inelastic behavior of the connection has been analysed by means of the Finite Element Simulation. The paper is well articulated, but I suggest the following remarks:
1. the authors must emphasize the novelty of the work.
2.Specify how the external diaphragms are fixed to the column.
3. In the finite element modelling, justify the choice of solid186 as an element of discretization.
4. At the end of lines 213, 235 there is not the final point of the sentence.
5. The authors should pay attention to the formatting of symbols within the manuscript
Author Response
Dear reviewer:
Thank you for your report and instructions regarding the review stage of our article “Numerical Analysis of the Seismic Behavior of a Steel Beam-to-Concrete-Filled Steel Tubular Column Connection Using External Diaphragms” by Christian Ramirez Ortiz, Albio D. Gutierrez Amador, and Jose L. Ramirez Duque to be consider for publication in Buildings, an MDPI Journal.
Please find our rebuttal letter in the attachment.

Reviewer 2 Report
This work presents a numerical model and an analysis of the inelastic behaviour under cyclic loads of a steel beam-to-concrete-filled steel tubular column connection using external diaphragms. I have a few questions and comments as follows:
Inline 15: the “numerical” model …
I wonder if those conclusions in lines 16 or 19 were obtained by authors or from previous studies or experiments?
Inline 32 SMFs appear for the first time and should be defined here. Not in the next paragraph.
I encourage authors to reduce the Introduction section. In my opinion, almost three pages of Introduction is somehow too long.
Authors should justify better the reason for doing this research and the novelty of the work.
Figures should be more precise—for example, Figure 8.
In Figure 10, the right-side Figure corresponds to the previous study. Hence it should be mentioned and cited that study.
I have a question: did the authors try to develop a new numerical model or simply they did simulate an experiment using Ansys and the predefined elements in ANSYS. Did they want to calibrate those predefined elements in ANSYS or what?
Did the authors suggest any further improvements to their work? Apparently, not!
Author Response

(The authors gave the same response as above.)

Reviewer 3 Report
Dear authors,
The work " Numerical Analysis of the Seismic Behavior of a Steel Beam-to-Concrete-Filled Steel Tubular Column Connection Using External Diaphragms" addresses a concrete issue and presents a FEM analysis with some merits. The paper is well-organized and the subject is worthy of investigation. The results are of practical importance. The following minor revisions are needed, please.
Please do not use citation pockets (e.g. [2–11]) but rather cite each reference individually showing why this reference has been cited (e.g. as described in [9]).
The words "Fpr" and "Mf" in lines 178 and 179 should be consistent with those in Figure 1.
Lines 195-198 and the titles in Figure 2 are repeated, so they need only be described once.
The description of the experiment, the description of the testing, and the counting seem to be clear and well prepared. Here, however, I see a fundamental problem that you have the same images as reference 19 - just add a reference to this article to describe the image.
There are some minor errors in lines 503-504 and 533-534. Please adjust.
Regards,
Author Response

(The authors gave the same response as above.)

Round 2
Reviewer 2 Report
-